# Exploring the Effect of Social Media and Group Chat Use on Social Isolation Among the Older Adults: A Study in Urban Japan

**DOI:** 10.3390/geriatrics10050131

**Published:** 2025-10-13

**Authors:** Yohei Sekikawa, Masafumi Kunishige, Taichi Hitomi, Kazumi Kikuchi

**Affiliations:** 1Department of Occupational Therapy, Faculty of Health Sciences, Kyorin University, Mitaka, Tokyo 181-8611, Japan; taichi-hitomi@ks.kyorin-u.ac.jp; 2Department of Occupational Therapy, Faculty of Health Science Technology, Bunkyo Gakuin University, Fujimino, Saitama 356-8533, Japan; mkunishige@bgu.ac.jp; 3Department of Occupational Therapy, Faculty of Health and Medical Science, Teikyo Heisei University, Toshima, Tokyo 170-8445, Japan; k228kazumi@thu.ac.jp

**Keywords:** social networking service, social media, group chat, social isolation

## Abstract

**Background:** Although research has been conducted on older adults and social media, the relationship between social media use and social isolation remains unclear. This study aimed to explore the relationship between social isolation and the frequency of use of social media and group chats. **Methods:** We measured social isolation using the Japanese version of the Lubben Social Network Scale (LSNS-6) in 411 older adults people living in urban areas. We used a questionnaire to survey their use of social networking services (SNS) such as LINE, Facebook, X (formerly Twitter), and Instagram, and their use of group chats. A separate questionnaire surveyed frequency of and participation in group chats. We analyzed associations between variables with logistic regression and a chi-squared test. **Results:** The most used service was LINE, with 51.3% of users participating in group chat. The analysis did not show an association between frequency of social media use and social isolation. However, group chat use, especially in groups of friends and acquaintances, was significantly negatively associated with social isolation (OR = 0.30, *p* < 0.001). **Conclusions:** This study revealed that LINE group chats may ameliorate social isolation among older adults. It also suggests that research focusing on its content and usage is needed.

## 1. Introduction

Social isolation refers to a situation in which there is objectively little interaction with others; it is often distinguished from subjective loneliness [1]. Social isolation has been associated with deteriorating health [2] and rising mortality [3,4,5], leading to solitary deaths and suicides. In fact, solitary deaths are increasing in urban areas of Japan [6], where people are prone to social isolation [7].

Social isolation has a large impact on the health of older adults [8]. Japan’s aging rate has reached 28.8% [9], and Japan, like the United Kingdom, has appointed a Minister of Loneliness to promote measures against social isolation and loneliness [10]. Social isolation is further exacerbated by new coronavirus infections [11,12], making its resolution an urgent issue. Online interactions, especially interactions on social media, may be useful in addressing this issue because social media enables interaction across time and distance.

Research has been conducted on the association between the use of online, membership-based social networking services (SNSs) and various health indicators among older adults. Social participation, well-being, and subjective views of health [13,14,15,16,17] are negatively correlated with social isolation [18]. Furthermore, a longitudinal study [19] focusing on social isolation is underway. However, some reports have questioned the relationship between social isolation and depression [20,21], and no clear conclusions can be drawn yet.

Prior reports suggest that the impact of social media use is related to the frequency of its use [18], the type of service [22], and the content of the service, rather than simple social media use. SNSs with group chat functionality have been shown to be beneficial for well-being [23,24] and social support [20], so they may also protect against social isolation. However, studies focusing on the frequency of social media use by service and the relationship between group chat and social isolation are limited, and there is a lack of research on social media use among the older adults, especially in Japan [25]. Therefore, this study aims to expand our knowledge by focusing on different types of SNSs and group chats among older adults living in urban areas of Japan. We hypothesized that the types of services used and the frequency of social media use are differently associated with social isolation (Hypothesis 1). We also hypothesized that group chat use is negatively associated with social isolation (Hypothesis 2). The results of this study provide insight into how social media use can mitigate social isolation among older adults.

## 2. Methods

### 2.1. Study Design

This study was conducted via a self-administered questionnaire survey from 15 November to 25 November 2022. Questionnaires were collected by mail and fax (questionnaires were mailed, and responses were collected by fax).

### 2.2. Procedure and Sample

The target population consisted of 411 randomly sampled older adults aged 65 years or older living in Japanese cities and their suburbs (Tokyo, Kanagawa, Chiba, and Saitama). Participants were identified using monitors registered with the Japan Management Association Research Institute: Tokyo (JMAR), a Japanese research firm. Participation in the study was voluntary and could be discontinued at any time without any reason or adverse effect. We explained in writing that the completed and submitted questionnaires would be used anonymously, in compliance with privacy rights.

### 2.3. Measurements

#### 2.3.1. Dependent Variables

Social Isolation: Social isolation was assessed using the Japanese version of the Lubben Social Network Scale (LSNS-6) [26]. This scale is a shortened version of the social network scale developed by Lubben [27] for older adults and consists of three questions assessing kinship and three questions assessing friendship. The three kinship questions are as follows: (1) “How many relatives do you see or contact at least once a month?” (2) “To how many relatives do you feel close enough that you could call on them for help?” (3) “With how many relatives do you feel sufficiently at ease that you can talk to them about private matters?” The friendship scale asks the same three questions, simply replacing ‘relatives’ with ‘friends’. The answers have the following options: 0 = none; 1 = one; 2 = two; 3 = three or four; 4 = five to eight people; 5 = nine or more people. The maximum score for the LSNS-6 is 30 points, with higher scores indicating a larger social network. Respondents scoring below 12 are considered socially isolated.

#### 2.3.2. Independent Variable

Definition of SNSs: In this study, we define an SNS as LINE, Facebook, X (formerly Twitter), and Instagram, which are widely used SNSs in Japan (Ministry of Internal Affairs and Communications).

LINE: LINE is the most widely used SNS in Japan. It allows for personal and group chats and calls, and it has similar characteristics to WhatsApp, Messenger (Facebook), and WeChat. It allows users to create groups with their preferred communities, such as family members or hobbyists, or to join existing community groups by invitation.

Definition of Social Media Use: For this study, social media use was defined as use of social media at least once a month in the past 2 months [21]. To determine social media use, we asked, “Are you currently using any of the services known as SNS, such as LINE, Facebook, X (Twitter), or Instagram?” The questionnaire included images and descriptions of typical SNS, such as LINE and Facebook, to convey the meaning of SNSs in an easy-to-understand manner. Those who answered “No” to this question were considered social media non-users.

Frequency of Social Media Use: Those who answered “Yes” to the aforementioned question were asked, “How often have you used SNS in the past two months?” Respondents were asked to choose one answer for each service. Responses included (1) less than once a month, (2) 1–3 times a month, (3) once a week, (4) several times a week, (5) every day, and (6) many times daily. Respondents who answered that they used SNSs less than once a month (option 1) for any of the services in this question were considered non-users for that service. For each SNS, those who used the service between one and three times a month (choice 2) were classified as low-frequency users, those who used it once to several times a week (choices 3 and 4) were classified as moderate-frequency users, and those who used it daily (choices 5 and 6) were classified as high-frequency users, based on Aarts et al. [21] (2015).

Group Chat Use: LINE users were asked, “Do you use group chat?” An image of the group chat and a description of the group chat were added to the questionnaire to explain the details.

Participation in Group Chats: Those who used group chats were asked to choose the type of group they participated in. The options were “(1) family/relatives, (2) friends/acquaintances, (3) hobbies/associations, and (4) people I met on SNS.” Respondents were asked to indicate all groups in which they participated.

#### 2.3.3. Covariates

To investigate the association between social isolation and social media use, we used sex, age, cohabitation, mental health, subjective health outlook, physical health, and frequency of going out as adjustment variables.

Mental Health: In this study, mental health was defined by depression status. The Geriatric Depression Scale—Japanese (GDS-S-J) [28], a shortened version of the depression scale for the older adults, was used to assess depression. The GDS-S-J is the Japanese version of the internationally widely used Geriatric Depression Scale—Short Version (GDS-15). In this study, a score of 6 or less was considered nondepressed, and 7 or more was considered depressed [28].

Subjective Health: Current subjective health perceptions were assessed using a four-item response scale of (1) best, (2) very good, (3) not so good, and (4) not good. Responses (1) and (2) were classified as good subjective health, and responses (3) and (4) were classified as poor subjective health. A four-point scale was adopted because previous studies have shown that even-numbered scales without a neutral midpoint may demonstrate higher reliability than odd-numbered scales [29].

Physical Health: The Kihon Check List (KCL) is a questionnaire developed by the Japanese Ministry of Health, Labour and Welfare to screen high-risk older adults who are likely to require nursing care. It is a simple assessment method that requires only a yes or no response to 25 questions. The higher the score, the more at risk the older adults are considered to be; the KCL has been used in Spain [30], Turkey [31], and China [32]. In this study, 20 of the 25 items were employed, excluding depression items. Respondents with a score of 5 or less were classified as being in good physical health, and those with a score of 6 or more were classified as being in frail physical health. We used these cutoffs because they have been found to be comparable to Fried’s criteria for frailty [33,34].

Frequency of Going Out: Those who went out less than once a week were classified as shut-ins [35].

### 2.4. Analyses

All statistical analyses were performed with EZR (version 1.61), which is a modified version of the R Commander software designed to add statistical functions frequently used in biostatistics. Statistical analyses for each hypothesis were conducted as follows.

**Hypothesis** **1.***Social isolation is associated with frequency of social media use among older adults*.

Logistic regression analysis was conducted using social isolation as the dependent variable, frequency of use of each SNS (LINE, Facebook, Instagram, and X) as the explanatory variable, and gender, age, cohabitation, depression, physical health, frequency of going out, and subjective health as adjustment variables.

**Hypothesis** **2.***Social isolation is associated with group chat use*.

Logistic regression analysis was conducted by adding group chat use to the model of Hypothesis 1 as an explanatory variable and removing the frequency of social media use. Additionally, a chi-squared test was conducted to examine the relationship between group participation and social isolation among group chat users.

To assess the validity and predictive performance of the logistic regression models examining SNS use and social isolation, we compared five models and evaluated multicollinearity (VIF), outlier influence (Cook’s distance), discriminative ability (ROC curve and AUC), calibration of predicted probabilities, and overall accuracy.

## 3. Results

Three hundred and fifty responses were obtained, of which five were excluded because of missing responses, leaving a total of three hundred and forty-five responses for analysis. Participant demographics are shown in Table 1. The average age of the analyzed participants was 74.0 ± 6.5 years (65–89); 52.8% (*n* = 182) were female, 87.5% (*n* = 302) were cohabiting, and 40.3% (*n* = 139) were socially isolated. Past surveys have shown that social isolation is in the range of approximately 10–20% [36] (20–30% during the pandemic period [37]), making this a relatively socially isolated group.

### 3.1. Rate and Frequency of Social Media Use

Table 2 shows the frequency of use of each SNS; among the SNSs, LINE had the highest usage rate. The breakdown of usage was 27.2% (*n* = 94) non-users, 6.4% (*n* = 22) low users, 19.1% (*n* = 66) medium users, and 47.2% (*n* = 163) high users, indicating that almost half of all respondents used LINE daily. However, many respondents were non-users of SNSs other than LINE; the rate of non-users for Facebook was 79.1% (*n* = 273), 84.1% for X (*n* = 290), and 86.4% for Instagram (*n* = 298). According to national surveys, social media nonuse rates among those in their 60s and 70s were 14.0–37.1% for LINE, 79.8–89.5% for Facebook, 79.0–88.8% for X, and 78.7–92.5% for Instagram [38]. This study confirmed the earlier findings, showing that the most used service is LINE and that social media use declines with age. Table 3 shows the utilization rate of group chats. Group chats, a function of LINE, were used by 51.3% (*n* = 177) of all respondents; among LINE users (*n* = 251), 70.5% used it. The most common group composition was family/relatives (*n* = 132), followed by friends/acquaintances (*n* = 108), hobbies/friends (*n* = 57), and nonacquaintances (*n* = 2).

### 3.2. Associations with Social Isolation

Table 4 shows the results of the logistic regression analysis with social isolation as the dependent variable. There was no relationship between the frequency of use of each SNS and social isolation. However, regardless of service, social isolation was significantly associated with being male and living alone, consistent with previous reports [39,40]. A significant association was also observed between group chat use and social isolation (OR = 0.30, 95% CI = 0.18–0.49, *p* < 0.001). To assess the predictive performance and validity of SNS usage models, five logistic regression models were constructed, each including a different SNS (LINE group, LINE, Facebook, X, Instagram). Diagnostic checks showed no issues with multicollinearity (Max VIF 1.398–1.407) or influential outliers (Max Cook’s D 0.0199–0.0259), as summarized in Section A.1 Table A1. Discrimination was highest for the LINE group model (AUC = 0.696), which also showed the highest classification accuracy (62.6%). Calibration analyses confirmed that predicted probabilities increased consistently with observed outcomes across all models, indicating that LINE group use is the most informative SNS variable for identifying social isolation. Hierarchical logistic regression analyses confirmed that sex and living arrangement remained significant predictors of social isolation after sequential adjustment for age, mental health, physical health, frequency of going out, and subjective health (Appendix A Table A2). Other covariates were not significantly associated, suggesting that sex and living together are independent predictors of social isolation among older adults.

Furthermore, there was a significant association between social isolation and participation in group chats with friends and acquaintances (*p* = 0.02) (Table 5).

## 4. Discussion

This study investigated the relationship between social isolation and social media use among older adults and tested hypotheses to explore measures to alleviate social isolation related to social media. Specifically, this study investigated the following hypotheses: social isolation is associated with different types of SNSs and frequency of use (Hypothesis 1), and social isolation is associated with the use of group chats (Hypothesis 2).

Among the older adult participants included in this study, social isolation was not associated with social media use, regardless of the type and frequency of social media use. Thus, we reject Hypothesis 1. Although there have been various reports on the relationship between social media use and social isolation, the results of this study indicate that social media use is not important for social isolation in urban areas in Japan. However, social isolation was significantly negatively associated with group chat use, supporting Hypothesis 2. No report has focused on group chat use and social isolation [41], so this study’s finding is new. The results also suggest that, among group chats, those that include friends and acquaintances are most important for social isolation. Various health indicators suggest that friend involvement is more important than family involvement [42,43], and our results support this.

Although group chats, a function of LINE, were associated with social isolation, the results did not indicate that overall LINE use was associated with social isolation, suggesting that it is not simply social media use but how one uses social media that is important for social isolation. Additionally, the characteristics of LINE, which is mainly used to communicate with known contacts, and the characteristics of group chats, which allow for communication with multiple contacts at once, suggest that being able to communicate with multiple known contacts may be important for social isolation.

Older adults are prone to a decrease in the number of friends and relatives with age [44,45], and there is a concomitant decrease in social interaction opportunities [46]. Engaging with multiple people through group chats may compensate for these decreases. The current results suggest that, even for those who go out infrequently because of physical or mental health constraints, being connected to a community on an SNS may deter social isolation.

Past reports have been inconclusive regarding the association between social isolation and social media use. This could be because the studies have not been conducted in a uniform manner, and, thus, there is a high degree of variability [47]. Another possible explanation is that different countries and cultures have different usage rates for social networking services. For example, it has been reported that Facebook usage is lower in Japan (Japan: 33%, USA: 82.9%) than in other countries (USA, UK, Germany) [48]. Because of the differences in services and utilization rates, caution must be exercised in drawing conclusions about the results of this study. Few reports have examined differences in social isolation among the older adults by type of SNS, with total social media use [21,22] sometimes being used as a measure. This study showed that it is not just the amount of use, but the way specific features are used that is important. There remains a need to investigate the characteristics of single services such as Facebook [18] and WhatsApp [49]. In addition, experimental studies are very limited [15], so more focused research is needed.

### Limitations

First, because this is a correlational study, a reverse causal relationship cannot be ruled out. It is possible that older adults with extensive social networks who are less prone to social isolation are more likely to use group chats. Therefore, there is a need to clarify the causal relationship through prospective research. In addition, the survey was limited to urban areas [50], where the Internet and smartphones are most prevalent in Japan. Therefore, the results may not be applicable to nonurban areas. Furthermore, SNSs vary from country to country and culture to culture, so the results cannot be generalized.

## 5. Conclusions

The association between social isolation and social media use among the older adults in this study did not differ by service. However, group chats may provide a means of helping older adults in urban areas, who are more likely to be socially isolated, to connect with society. Additionally, it is important to join groups of friends and acquaintances with whom one can interact simultaneously, rather than simply participating in a particular SNS.

## Figures and Tables

**Table 1 geriatrics-10-00131-t001:** Descriptive characteristics of the study population.

Age, Mean (*SD*)	74.0 (6.5), Range 65–89
Female	52.8% (*n* = 182)
Living alone	12.5% (*n* = 43)
Depressive symptoms	14.5% (*n* = 50)
Social isolation	40.3% (*n* = 139)
Subjective health, bad	36.2% (*n* = 125)
Physical health, frail	27.5% (*n* = 95)
Frequency of going out, shut-in	5.2% (*n* = 18)

Cronbach’s α coefficients were 0.79 (95% CI: 0.76–0.82) for depressive symptoms (GDS), 0.85 (95% CI: 0.82–0.87) for social isolation—family, and 0.84 (95% CI: 0.81–0.86) for social isolation—friends (LSNS-6).

**Table 2 geriatrics-10-00131-t002:** Frequency of use of each social networking service.

Frequency of Use	LINE (*n*)	Facebook (*n*)	X (Twitter) (*n*)	Instagram (*n*)
Non-use	27.2% (94)	79.1% (273)	84.1% (290)	86.4% (298)
Low	6.4% (22)	3.8% (13)	4.6% (16)	3.5% (12)
Moderate	19.1% (66)	9.9% (34)	4.9% (17)	5.8% (20)
High	47.2% (163)	7.2% (25)	6.4% (22)	4.3% (15)

**Table 3 geriatrics-10-00131-t003:** Percentage of group chat usage.

Frequency of Use	Group Chat Non-Use *n* = 168 (*n*)	Group Chat Use *n* = 177 (*n*)
Non-use	27.2% (94)	0% (0)
Low	4.4% (15)	2.0% (7)
Moderate	7.5% (26)	11.6% (40)
High	9.6% (33)	37.7% (130)

**Table 4 geriatrics-10-00131-t004:** Results of logistic regression analysis of the association between SNS use and social isolation.

	LINE		Facebook		X		Instagram		Group Chat	
Variable	OR (95%CI)	*p*	OR (95%CI)	*p*	OR (95%CI)	*p*	OR (95%CI)	*p*	OR (95%CI)	*p*
Sex	1.82 (1.15–2.88)	*	1.93 (1.23–3.04)	**	1.89 (1.21–2.97)	**	1.89 (1.20–2.96)	**	1.86 (1.16–2.97)	**
Age	1.02 (0.98–1.06)	0.43	1.02 (0.98–1.06)	0.27	1.02 (0.99–1.06)	0.25	1.02 (0.98–1.06)	0.29	0.99 (0.95–1.03)	0.59
Living together	0.39 (0.20–0.77)	*	0.38 (0.19–0.75)	**	0.39 (0.20–0.77)	**	0.38 (0.19–0.75)	**	0.36 (0.18–0.74)	**
Mental health	1.11 (0.57–2.16)	0.75	1.13 (0.58–2.19)	0.72	1.12 (0.58–2.18)	0.73	1.12 (0.58–2.18)	0.73	1.1 (0.55–2.20)	0.78
Subjective health	0.7 (0.42–1.18)	0.19	0.71 (0.42–1.20)	0.20	0.71 (0.42–1.20)	0.20	0.72 (0.43–1.21)	0.21	0.73 (0.42–1.24)	0.24
Physical health	0.71 (0.39–1.28)	0.25	0.7 (0.39–1.26)	0.23	0.71 (0.39–1.27)	0.24	0.7 (0.39–1.26)	0.24	0.69 (0.38–1.27)	0.24
Frequency of going out	2.06 (0.74–5.74)	0.17	2.14 (0.76–6.01)	0.15	2.11 (0.75–5.93)	0.16	2.24 (0.78–6.44)	0.13	2.52 (0.88–7.23)	0.09
Frequency of use	0.92 (0.76–1.11)	0.38	0.89 (0.70–1.14)	0.36	0.98 (0.75–1.28)	0.87	0.91 (0.66–1.24)	0.53		
Group chat use									0.3 (0.18–0.49)	***

OR = odds ratio. CI = confidence interval. * < 0.05, ** < 0.01, *** < 0.001; sex (0 = female; 1 = male); living together (0 = alone; 1 = together); mental health (0 = good; 1 = depression); subjective health (0 = bad; 1 = good); physical health (0 = good; 1 = frail); frequency of going out (0 = not a shut-in; 1 = shut-in); group chat use (0 = non-use; 1 = use).

**Table 5 geriatrics-10-00131-t005:** Results of chi-squared test of group chat participation and social isolation.

Group	Participation	Not Social Isolation	Social Isolation	χ2	*p*-Value
Family/Relative	No	29	16	2.17	0.14
	Yes	100	32		
Friends/ Acquaintances	No	44	25	4.75	0.02 *
	Yes	85	23		
Hobbies/ Associations	No	87	33	0.02	0.86
	Yes	42	15		
Non-acquaintances	No	127	48	0.75	0.38
	Yes	2	0		

Group chat users (*n* = 177); * *p* < 0.05.

## Data Availability

The data presented in this study are available on request from the corresponding author due to ethical and privacy considerations.

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
