# Peer review of "Exploring the Effect of Social Media and Group Chat Use on Social Isolation Among the Older Adults: A Study in Urban Japan"

_geriatrics, 2025, doi:10.3390/geriatrics10050131_

Round 1

Reviewer 1 Report

Comments and Suggestions for Authors

This study is a useful addition to the literature on social isolation in older adults. The study methodology is appropriate and clear.  The conclusions are supported by the results. I appreciate the important limitation noted by the authors regarding the possibility of reverse causality.

I note that only 12.5% of the participants lived alone which is less apparently than the number of the older population in Japan (19.4% in 2021 according to one report).  It would be most interesting to do a similar study focused only on those living alone and therefore most likely to be isolated.

A couple of minor suggestions re wording:

On line 207 suggest "related to social media" rather than "caused by social media".

On line 215 - add "negatively"  associated etc.

Thanks for the opportunity to review this interesting study.

Reviewer 2 Report

Comments and Suggestions for Authors
  1. For the content related to the research purpose and hypotheses in lines 56 to 60, it is recommended to clearly define its targeted nature for the elderly population.
  2. It is recommended to report the reliability and validity evidence of different parts of the questionnaire (mature scale and self-developed scale) in your current research separately.
  3. The article lacks the model diagnosis section. It is suggested that you describe in the method what diagnoses you have made for the logistic regression model (Linearity, Multicollinearity, Outliers & Influential Points, etc.) Provide an assessment of its Discrimination and Calibration, etc., and present the key results and conclusions of the test (VIF value table, residual plot, influence plot, sensitivity analysis results, etc.) in the results or appendices.
  4. It is suggested that corresponding hierarchical analysis be given to "social isolation was significantly associated with being male and living alone." in lines 190-191.

Round 2

Reviewer 2 Report

Comments and Suggestions for Authors

The validity of the GDS-S-J scale and the LSNA-6 scale you used can be referenced from previous studies, but reliability varies by sample. It is recommended that you re-report internal reliability in your own sample (for example, Cronbach's α).
